



# Obtaining fatigue-based frequency domain specifications for the design of controllers in wind turbines

Irene Miquelez-Madariaga[1], Jesús Arellano[2,1], Daniel Lacheta-Lecumberri[3,1], and Jorge Elso[1]

[1]Universidad Pública de Navarra
[2]Siemens Gamesa Renewable Energy
[3]VORS Control

**Correspondence:** Irene Miquelez-Madariaga (irene.miquelez@unavarra.es)

**Abstract.** This work presents a methodology to (i) estimate fatigue using linear models and (ii) generate control specifications directly linked to the mechanical fatigue caused by driving loads for a wind turbine applications. The method is intended for frequency domain controller design techniques such as QFT or $H_\infty$ and is based on Dirlik's method for fatigue assessment. The main advantage of using frequency domain approach is that the need of computationally expensive processess such as

5    the generation of turbulent wind fields or aeroelastic simulations is reduced. As a consequence, the controller design method becomes more agile. The method has been validated by designing controllers for the reference 15MW wind turbine based on fatigue specifications, obtaining simulation results with OpenFAST and comparing the fatigue results from a rainflow algorithm with the linear estimation. Fatigue has been reduced by 22% to 35% at different wind speeds corresponding to above rated operation. The mean fatigue estimation error is 1.07%, proving the method is suitable for a wind turbine control application.

## 1 Introduction

The design of wind turbines is an inherently multiobjective problem. In order to reduce the Levelized Cost of Energy (LCoE, Bruck (2018)), wind turbines should be designed to maximize the generated power and to minimize design, production and operation expenses. When the design focuses specifically on the control strategies, these main objectives are translated into maximizing generated power, regulating the generator speed, and reducing the driving mechanical loads.

Even though the design of controllers for wind turbines is a well-known problem in the academic literature and there exist plenty of different proposals, frequency domain methods stand out due to some advantageous characteristics (Song (2022); Singh (2016)). The first one is the fact that experimental models of the system are obtained by modal identification or similar techniques. Therefore, they are directly represented in the frequency domain and can be used with little post-processing. The second advantage, which might be the most relevant one, is that wind theoretical models are described by means of the power

spectrum (Kaimal (1972); Mann (1998)). Then, with the three main elements of the design process -system, controller, and disturbance- defined in the frequency domain, computationally-heavy simulations are not required, making the design process agile and allowing multiple iterations.

   In this context, designing controllers that ensure a reduction of the mechanical fatigue can be a challenge. According to the standard (IEC (2020)), fatigue must be assessed by means of aeroelastic simulation of turbulent wind fields. Both the generation





of the wind fields and their simulation are time-consuming steps that are incompatible with a swift workflow. Besides, in order to get the fatigue indicator, load cycles are quantified using the nonlinear rainflow counting algorithm and the Wöhler curve.

All in all, a gap exists between the controller design environment, which uses linear frequency-domain models, and the evaluation environment which requires nonlinear time-domain simulations. Some work has been done to solve this problem for a general wind turbine design application (Tibaldi (2016)). This work proposes a link between the design and the evaluation

environments by providing a method to establish controller design specifications that are set in the frequency domain and are at the same time related to mechanical fatigue. The proposed method has two main characteristics: (1) it allows to identify the range of frequencies in which the specification contributes more to the overall fatigue and (2) it allows to quantify the expected fatigue decrease when a change in the specification or the controller is introduced.

In Section 2, a context on fatigue evaluation is presented, focusing on frequency domain methods. Then, in Section 3 the

main contributions of this work are described, namely a method for obtaining the sensitivity of the damage equivalent load to changes in the signal spectrum and a way to propose control specifications to reduce fatigue. Section 4 provides a validation of the method based on the design of controllers for a 15MW wind turbine. Lastly, Section 5 summarizes the main results and conclusions of the work.

## 2   Frequency domain assessment of mechanical fatigue

The IEC 61400 standard for the design of wind turbines (IEC (2020)) states that the mechanical damage caused by fatigue in a specific wind turbine configuration has to be evaluated by the simulation of turbulent wind fields at different operating points using an aeroelastic model of the wind turbine. The evaluation process established by the standard starts by simulating the operation of the wind turbine in order to obtain realistic time series of the mechanical loads. Then, the number and amplitude of the load cycles are computed using a rainflow-counting algorithm. Finally, the total accumulated damage is obtained by

using the Wöhler curve (**?**) and a Palmgren–Miner linear damage hypothesis (Manson (1994)).

Although this process is compulsory for the certification of a wind turbine configuration, it might not be the most suitable approach for the current controller's design process, for various reasons. Firstly, mechanical damage does not have an explicit relation with most of the design parameters, which means that the effect of a change in the design is unknown until tested in simulation. Additionally, the generation of wind fields and their simulation in the aeroelastic model are time-consuming steps

that prevent the design process from being agile, specially due to its iterative nature.

The literature on fatigue assessment includes a set of methods that approximate the fatigue with expresions that depend on the spectral properties of the load. The most basic approximation is the narrow-band method (Wirsching (1980)), an analytic method that provides a framework for estimating the fatigue damage in structures subjected to random stress processes. The method assumes that the stress process is narrow-band, thus simplifying the analysis by treating the stress cycles as approxi-

mately sinusoidal with a constant amplitude and frequency. Besides, the narrow-band method draws an analogy to the rainflow counting method used in the time domain. It estimates the number of stress cycles and their respective amplitudes using the





properties of the narrow-band random process as

$$\Delta D_{NB} = \nu_0 N_0^{-1} S_0^{-k} (2m_0)^{\frac{k}{2}} \Gamma\left(1 + \frac{k}{2}\right), \tag{1}$$

where $\nu_0$ is the zero-crossing rate, $N_0$ is a normalization factor related to the number of cycles, $S_0$ is the spectral width parameter, $m_0$ is the zeroth spectral moment, $\Gamma()$ is the Gamma function and $k$ is a material exponent of the S-N curve.

Most of the existing methods offer variations on the narrow-band method to offer more precise results for wide-band processes (Benasciutti (2005, 2012)). They often follow an empirical approach and are therefore precise when the analyzed load is similar to the training data. Among the empiricals solutions to frequency-domain fatigue assessment, Dirlik's work (Dirlik (1985)) is one of the most accurate and widely accepted methods for estimating fatigue damage under random loading conditions. It is specifically designed to handle a wide range of frequency contents. This method provides an efficient and accurate approach to predict fatigue life by leveraging the statistical properties of the stress response. Dirlik uses a probabilistic approach to estimate the distribution of stress cycles, deriving an empirical formula for the probability density function (PDF) of stress ranges in a random loading process. Essentially, it uses the spectral properties of the load and the mechanical properties that define the Wöhler-Curve. The formula accounts for the distribution of peaks and valleys in the stress history, providing a detailed statistical description. The empirical formula for the PDF is given by

$$\Delta D_{Dirlik} = \nu_p N_0^{-1} S_0^{-k} (m_0)^{\frac{k}{2}} \left[ G_1 Q^k \Gamma(1+k) + 2^{\frac{k}{2}} \Gamma(1 + \frac{k}{2})(G_2|R|^k + G_3) \right], \tag{2}$$

where $G_1$, $G_2$, and $G_3$ are empirical constants derived from fitting the model to experimental data, $Q$ is a parameter related to the peak factor, which is derived from the ratio of higher-order spectral moments and characterizes the non-Gaussianity of the stress process and $R$ is the mean value of the stress process normalized by its standard deviation, describing the relative location of the process mean with respect to the stress amplitude.

## 3 Using linear models for the estimation of fatigue and the generation of control specifications

The main input to the frequency-domain fatigue estimation methods presented in Section 2 is the power spectrum of the evaluated load, which can either be estimated from simulation or measured data or inferred from theoretical models. Even though the use of simulation data provides closer results to time-domain evaluation, the use of theoretical models narrows the gap between controller design and fatigue estimation.

Assuming a linear model of the open loop plant $G_{y,u}(i\omega)$, where $u$ and $y$ denote the input and output signal respectively, the closed loop transfer function $T_{y,u}(i\omega)$ is a function of $G_{y,u}(i\omega)$ and the controller $C(i\omega)$. As an example, the effect of wind $W(i\omega)$ on the generator speed $\Omega_g(i\omega)$ in the controlled system is

$$T_{\Omega_g,W}(i\omega) = \frac{G_{\Omega_g,W}(i\omega)}{1 + G_{\Omega_g,\beta}(i\omega)C(i\omega)}. \tag{3}$$

Similar relations can be obtained for any input-output combination in the system.





If the closed loop transfer function is known, the theoretical auto-spectrum of the output is obtained using the relation

$$S_{yy}(\omega) = |T_{y,u}(i\omega)|^2 S_{uu}(\omega), \tag{4}$$

where $S_{uu}(\omega)$ is the auto-spectrum of the disturbance input. In the case of wind turbine control, the disturbance the disturbance input is the wind, as well as the waves for offshore wind turbines. Typically, the rotor effective wind speed (or disk average wind speed) is used together with linear models. Theoretical models of rotor effective wind speed are obtained from the combination of turbulence models at a single point, such as Kaimal spectrum, and the spatial coherence models described in the standard. Then, once the output spectum is known, fatigue estimation methods such as Dirlik's can be used to assess fatigue.

With this information in hand, a new method for the prediction of fatigue using linear models is proposed. This method has two main goals: (i) detect the frequencies where the load has a greater contribution to fatigue and design according control specifications and (ii) estimate the change in fatigue for a new control configuration so that the number of nonlinear simulations during the design of a satisfactory controller is reduced. The method entails the following steps:

1. An initial control configuration is assumed, either a baseline controller that needs to be improved or an open loop plant. The initial controller $C_0(i\omega)$ (which could be $C_0(i\omega) = 0$) can be simulated to obtain an initial output spectrum $S_{yy,0}(\omega)$. Simulation outputs are used to obtain an initial fatigue evaluation $D_0$.

2. The amplitude of the spectrum at a single frequency $\omega_0$ is reduced by a 1%, generating a new spectrum $S_{yy,\omega_0}(i\omega)$.

3. A new fatigue estimation $D_{\omega_0}$ is obtained for $S_{yy,\omega_0}(i\omega)$.

4. The fatigue sensitivity at a single frequency $\Delta D_\%(\omega)$ is calculated as

$$\Delta D_\%(\omega_0) = \frac{D_0 - D_{\omega_0}}{D_0} \cdot 100. \tag{5}$$

5. The process is repeated for the whole range of frequencies in which the spectum is defined, resulting in $\Delta D_\%(\omega)$, a frequency dependent function that represents the effect of the each frequency component of the spectrum on the total fatigue.

Figure 1 shows an example of the proposed method. In the upper plot, a linear model of the system is represented by means of a magnitude Bode plot. The middle plot represents the load power spectrum obtained from simulation data. Lastly, the lower plot contains the fatigue sensitivity obtained as previously described.



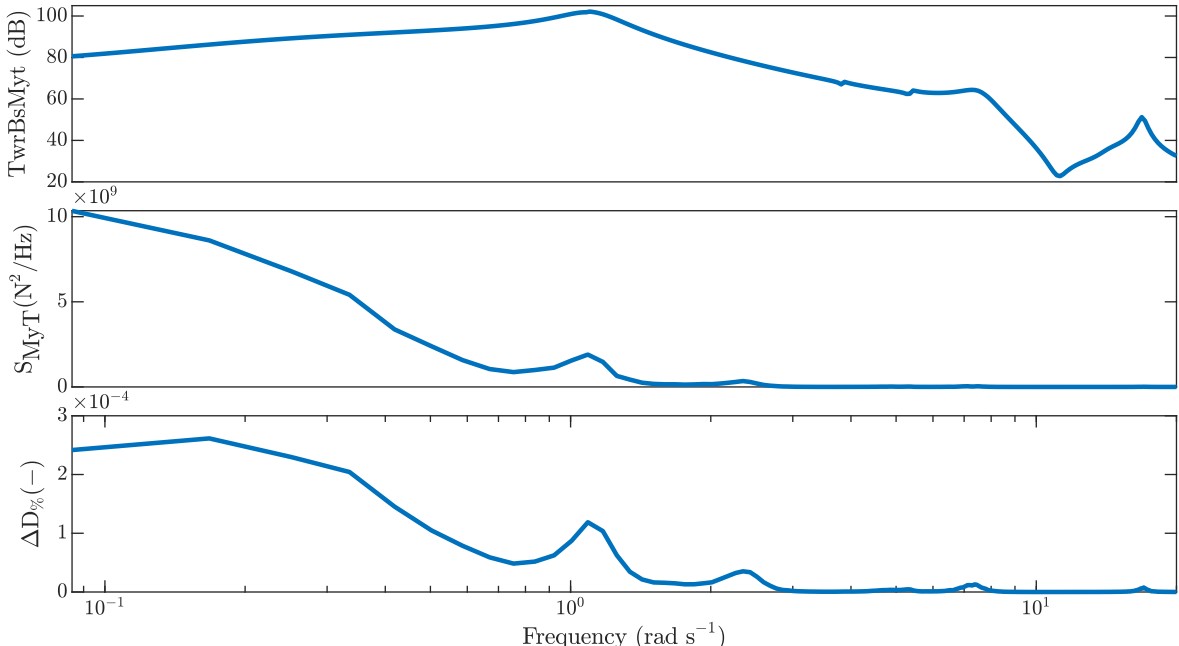

**Figure 1.** This figure shows different aspects of the relation between wind and tower base load. The upper plot shows the magnitude of the closed loop transfer function between wind and load. The middle plot, that represents the load power spectrum, shows how the biggest amplitudes of the load appear at the lower frequencies. The lower plot, which represents how a change in each frequency of the load would affect to fatigue, shows that higher frequencies lead to higher fatigue.

Many controller design techniques start by proposing a desired response for the controlled systems. This proposal, also known as control specification, can be in the form of parameters of a standard response or, as is the case for QFT or $H_\infty$, as an objective frequency response that can be denoted as $W_y$. By applying Equation 4, a theoretical output spectrum of the desired response $S_{W_y}(\omega)$ can be obtained. The achieved attenuation with respect to the baseline controller can be calculated as

$$\Delta S_{W_y}(\omega) = \frac{S_{yy,0}(\omega) - S_{W_y}(\omega)}{S_{yy,0}(\omega)} \cdot 100. \tag{6}$$

Under the assumption of linearity, which has been tested via simulation, the fatigue variation obtained with the control specification can be calculated as

$$\Delta D_{W_y} = \sum_\omega \Delta D_\%(\omega) \cdot \Delta S_{W_y}(\omega), \tag{7}$$

and the actual fatigue would be

$$D_{W_y} = \sum_\omega \Delta D_\%(\omega) \cdot \Delta S_{W_y}(\omega) D_0. \tag{8}$$

Using again Equation 4, Equation 7 can be rewritten as

$$\Delta D_{W_y} = \sum_\omega \Delta D_\%(\omega) \cdot \frac{|T_{yu,i}(i\omega)|^2 - |W_y(i\omega)|^2}{|T_{yu,i}(i\omega)|^2}, \tag{9}$$





where $\Delta D_\%(\omega)$ depends on the initial set of simulations and $\frac{|T_{yu,i}(i\omega)|^2 - |W_y(i\omega)|^2}{|T_{yu,i}(i\omega)|^2}$ depends exclusively on linear models. With this result, the second objective for the method has been achieved.

## 4 Validation for the 15MW reference wind turbine

The validation of the method proposed in this work is performed by using QFT for the design of linear controllers based on the fatigue specifications. The performance of the controllers is evaluated by their simulation in aeroelastic code (OpenFAST (2024)) and the post processing of the resulting time series with a rainflow counting algorithm. For that purpose, the 15MW reference wind turbine with the ROSCO controller (Abbas (2022)) is used.

### 4.1 Sistem description

The wind turbine model used for this study is the IEA 15MW reference wind turbine (Gaertner (2020)). This turbine is an offshore, monopile model with three blades and a horizontal axis. The most relevant parameters of the model are gathered in Table 1.

| Parameter | Value |
| --- | --- |
| Rotor diameter | 240 m |
| Hub height | 150 m |
| Cut-in rotor speed | 5 rpm |
| Rated rotor speed | 7.56 rpm |

**Table 1.** Value of the main parameters of the 15MW reference wind turbine.

Initially, this wind turbine operates using the reference Open-Source Controller (ROSCO) as a baseline control, which was developed with the aim of offering a modular control structure, with an industry-level performance and compatibility with the OpenFAST design and simulation environment. ROSCO includes the control strategies corresponding to the main operating regions of a wind turbine, from low speed winds (Region 1) to above rated wind speed (Region 3). There are two main control strategies, corresponding to pitch and torque controllers. Torque control is especially relevant at lower wind speeds, in which a quadratic control law is used to ensure maximal power production. Pitch control is used at above rated wind speeds to ensure a constant generator speed and nominal power production. Besides, ROSCO includes switching, filtering and load reduction strategies.

For the purpose of this work, which is linear fatigue assessment, the analysis focuses on above rated operation. There, the control strategy consists in a set of linear controllers with scheduled gain to face the nonlinear dynamics of the wind turbine. Typically, these controllers follow a PI structure in which the integrator rejects the effect of wind in the lower frequencies and the zero increases the phase margin of the system. Figure 2 shows the magnitude plot of the PI controller operating at a 19m/s




wind speed (lower plot) and its effect on the generator speed and tower base load. The main effect of the controller can be appreciated in the generator speed plot. There, the red line correspoding to ROSCO controller shows the attenuation of the effect of wind on the generator speed at the lower frequencies.

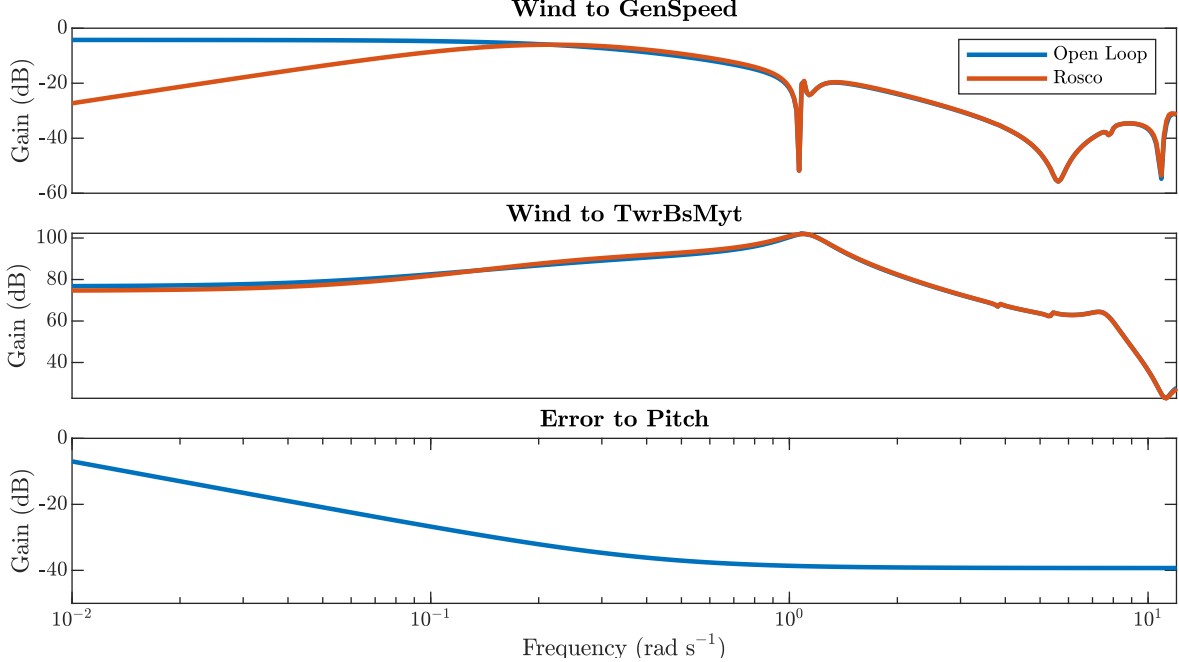

**Figure 2.** Effect of the ROSCO controller on the dynamics of the system. The upper plot shows the effect of the PI controller on the generator speed, which is more relevant at the lower frequencies. The middle plot shows the effect of the controller on the tower base bending moment. Lastly, the lower plot shows the magnitude of the feedback controller.

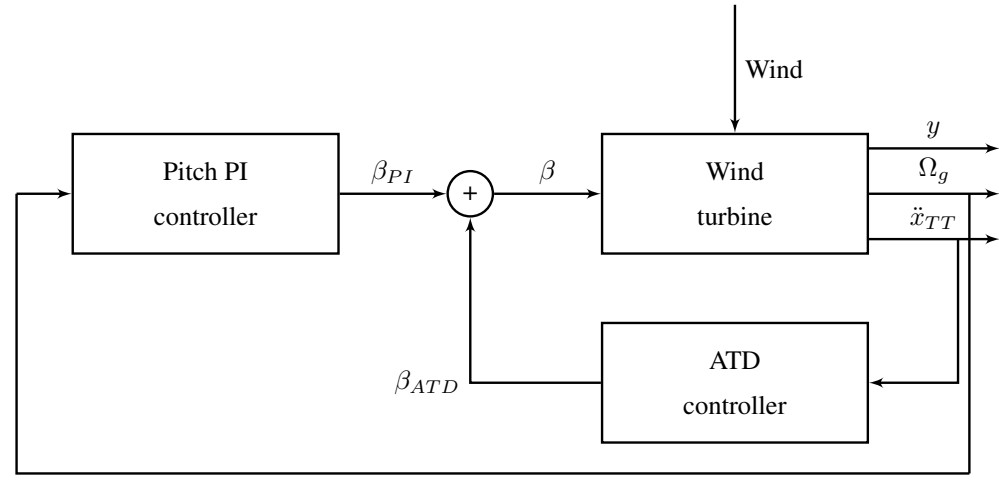

**Figure 3.** Block diagram of the control structure.



## 4.2 QFT fundamentals

Quantitative feedback theory is a controller design methodology that allows to obtain robust, multivariable and multiobjective solution (Elso (2017)). The main steps in the design process are:

1. Obtaining the model of the plant. QFT is based on the use of linear, frequency domain models of the system. Robustness is achieved by using uncertain models during the design.

2. Selecting the design frequencies. QFT design is performed on a discrete set of frequencies as reference. These frequencies should cover all the relevant dynamics of the system.

3. Choosing the specifications. In the context of QFT design for win turbine controllers, control specifications are typically upper bounds on the closed-loop disturbance rejection function of the different control objectives. These upper bounds can be constant values or transfer functions and should ideally be linked to performance indicators such as the generator speed standard deviation or the mechanical fatigue of driving components. Besides, a stability specification is imposed as an upper bound on the complementary sensitivity function.

4. Generation of bounds. At each design frequency, the regions of all controllers meeting each specification is found in the complex plane. The boundary of these regions, multiplied by the nominal plant, in the complex plan is called bound and there exists one per specification and design frequency.

5. Controller synthesis, also known as loopshaping, consists tuning the controller parameters until the open-loop transfer function that lies within the bounds at every design frequency.

6. Checking the specifications. The first validation step uses the uncertain linear model of the plant to ensure that specifications are met at every frequency and not only the design ones.

7. Simulation. If the linear uncertain model of the plan has been derived from a more complex mode (i.e. a nonlinear one), as is the case for wind turbines, a second evaluation of the performance of the controllers is performed via simulation.

The whole process is carried out with the aid of the QFT Toolbox (Yaniv (1997)), which includes graphical tools for the design of specifications and the loopshaping. QFT is inherently an iterative method, more so when several controllers are being tuned at the same time, and requires some skills in the loopshaping stage. As a counterpart, it is a versatile and transparent methodology, that grants engineers a full control of the design process.

## 4.3 Controller design

The design of controllers starts with the design of the control specifications, which follows the procedure presented in Section 4. The first step consists on obtaining a set of simulations of the system with the baseline control. In particular, four wind seeds with length 600s have been simulated at each operating point. Then, these simulations are used to calculate the output spectra and an initial fatigue evaluation. Lastly, the variation of fatigue caused by a change in the tower base load spectrum





is calculated based on the simulation results. Figure 1 shows the relation between the magnitude Bode plot, the spectrum of the load and the variation of fatigue for the simulations corresponding to a 19m/s mean wind speed. While the Bode plot only holds information on the response of the system, the spectrum of the load includes information on the disturbance, showing a higher amplitude in the lower frequencies. Besides, as the spectrum has been obtained from simulation data, a peak can be observed at around 2.3 rad/s, that corresponds to the 3P frequency. Lastly, the lower plot shows how the lower frequencies have the greater impact on fatigue, but the peaks corresponding to the first fore-aft mode of the tower and the 3P frequency are also relatively relevant.

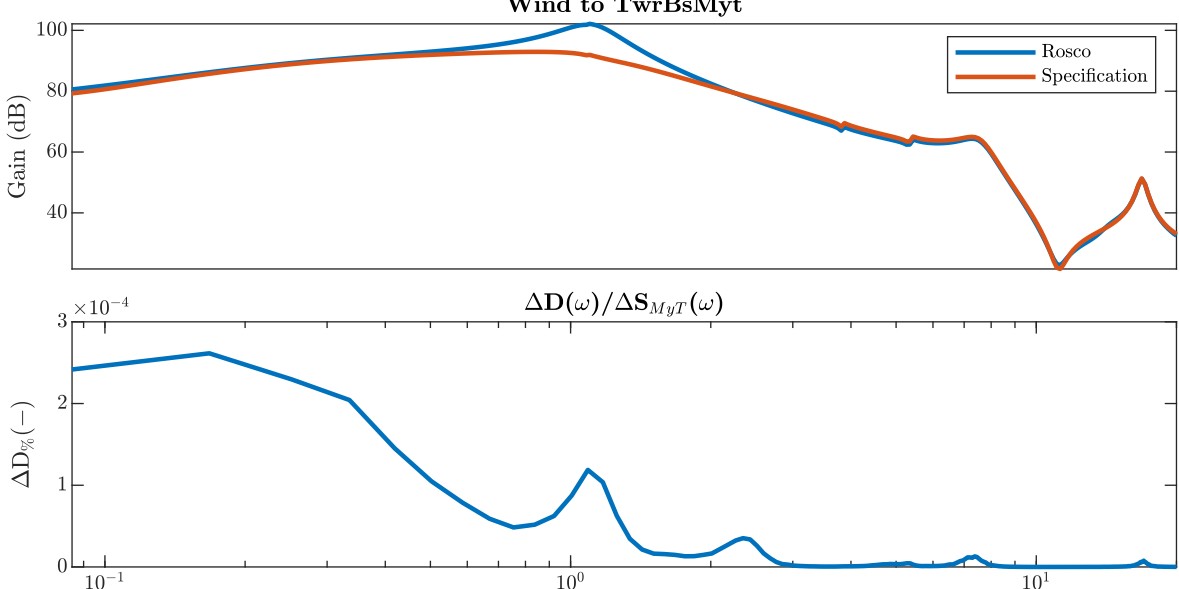

**Figure 4.** Design of the control specification for an operating point corresponding to a 19 m/s wind speed. The control specification has a similar magnitude as the ROSCO controller at all frequencies but the one sorrounding the first fore aft mode of the tower (1.1 rad/s).

The design of the specification $W_y$, which takes as a starting point the closed loop transfer function produced by the baseline controller, is based on the information provided by function $\Delta D_\%(\omega)$. Even though the lower frequencies have the greatest impact on the fatigue (see the lower plot in Figure 4), the designer must keep in mind that the main control objective is to have a constant power production, for which the lower frequencies are key. As a consequence, the control effort is located around the first fore-aft mode of the tower, which appears at approximately 1.1 rad/s. The control specification has been obtained with the aid of the *lpshape* function of the *QFT Toolbox*. A set of notch filters have been used to reduce the magnitude of the specifications at the chosen frequencies until the desired reduction of fatigue has been obtained.

As already mentioned, power production and, in turn, generator speed regulation are the most relevant objectives in the design of the pitch controllers. As a consequence, they must also be included in the design specifications. In this case, the closed loop transfer function (wind to generator speed) obtained with the ROSCO controller is used as specification. This way, a better performance than the baseline controller is expected in terms of generator speed regulation.

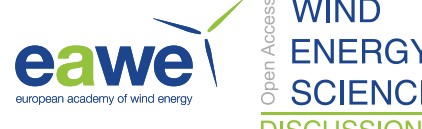

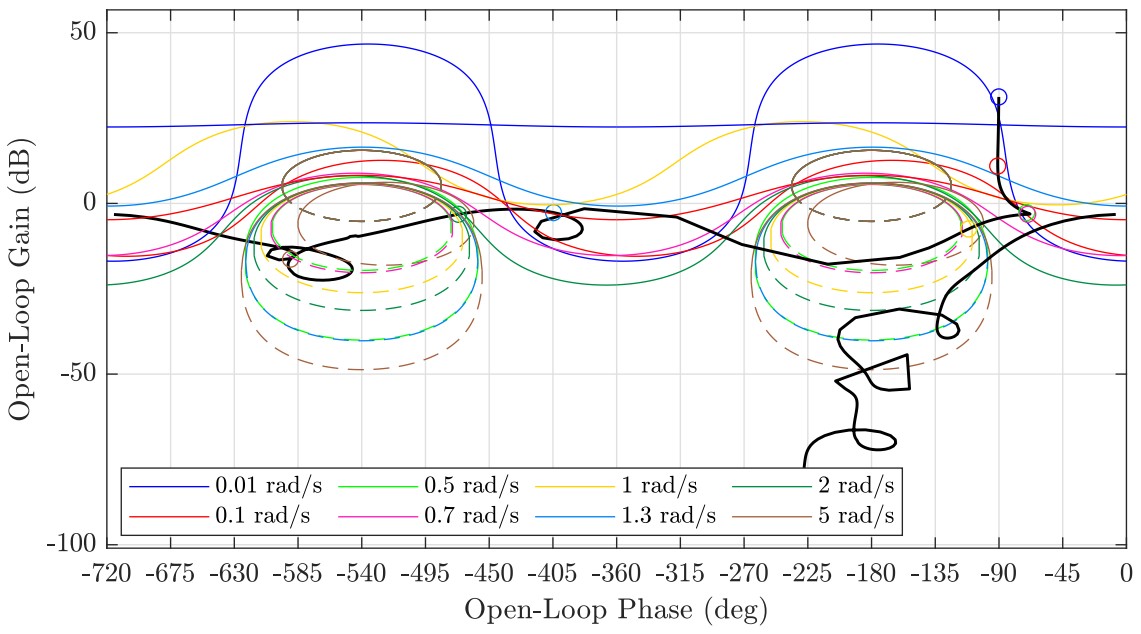

**Figure 5.** Bounds for the loopshaping of the main feedback pitch controller. Then main characteristic of this controller is the presence of an integrator to eliminate steady-state error and the use of low frequency poles to increase magnitude and phase below 1 rad/s. The open-loop nominal transfer function meets all bound at all frequencies.

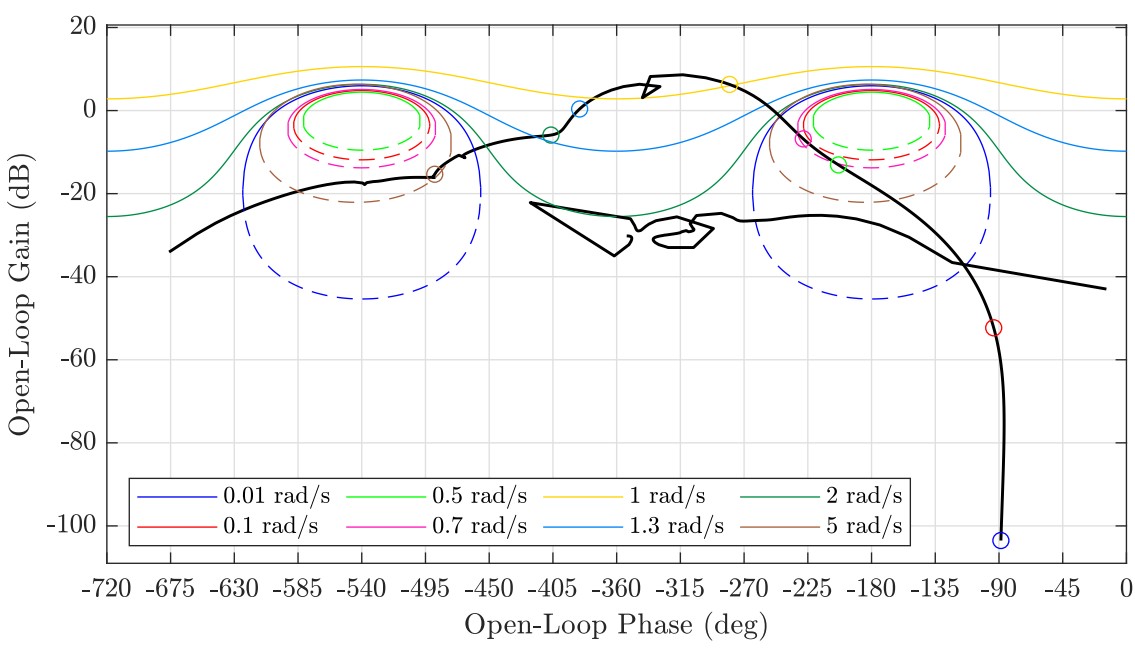





**Figure 6.** Bounds for the loopshaping of the ATD controller. This controller has a pole 0.3276 rad/s and a zero at 4.357 rad/s. The main control effort of the ATD strategy appears close to 1rad/s, at the first fore-aft mode of the tower.

Once the design specifications for the two control objectives have been obtained, the bounds for the controllers have been obtained using the procedure described in Section 5.2. The bounds are obtained using function *genbounds* and the design of controllers is performed with the aid of *lpshape*. As two different controllers have been designed for each operating point -the feedback pitch controller and the ATD- the design process is iterative. Due to the lack of ATD controller in the baseline configuration, the design process begins with the tuning of a preliminary ATD controller. Then, several iterations are performed until specifications are met.

Figures 5 and 6 show the final design of the pitch and ATD controllers for the operating point corresponding to a 19 m/s average wind speed. At this wind speed, the pitch controller is

$$C_\beta(s) = \frac{-4.8167(s+0.6)(s^2+0.7337s+0.2441)}{s(s^2+1.872s+2.306)(s^2+2.297s+26.67)}. \tag{10}$$

The ATD controller at the same operating point is

$$C_{ATD}(s) = \frac{0.014187(s+4.357)}{s+0.3276}. \tag{11}$$

The bound are met tightly at frequencies sorrounding 1 rad/s and with a greater margin for lower frequencies, as seen in Figures 5 and 6. A similar conclusion can be inferred from the Bode plots presented in Figure 7. There, the stars represent the design frequencies, which are used for the calculation of bounds and the loop shaping. As already anticipated, at frequencies below 1rad/s, either one or both specifications are met with some margin. At higher frequencies, the tower base load equals exactly the specification. With a simple enough control structure, a good result is expected in between design frequencies. Figure 7 shows that the QFT controller provides a poorer response than the specification between 0.8 and 1 rad/s and between 2 and 5 rad/s for the tower base load. However, the overal response is better than the specification and shows a significant improvement with respecto to ROSCO controller.

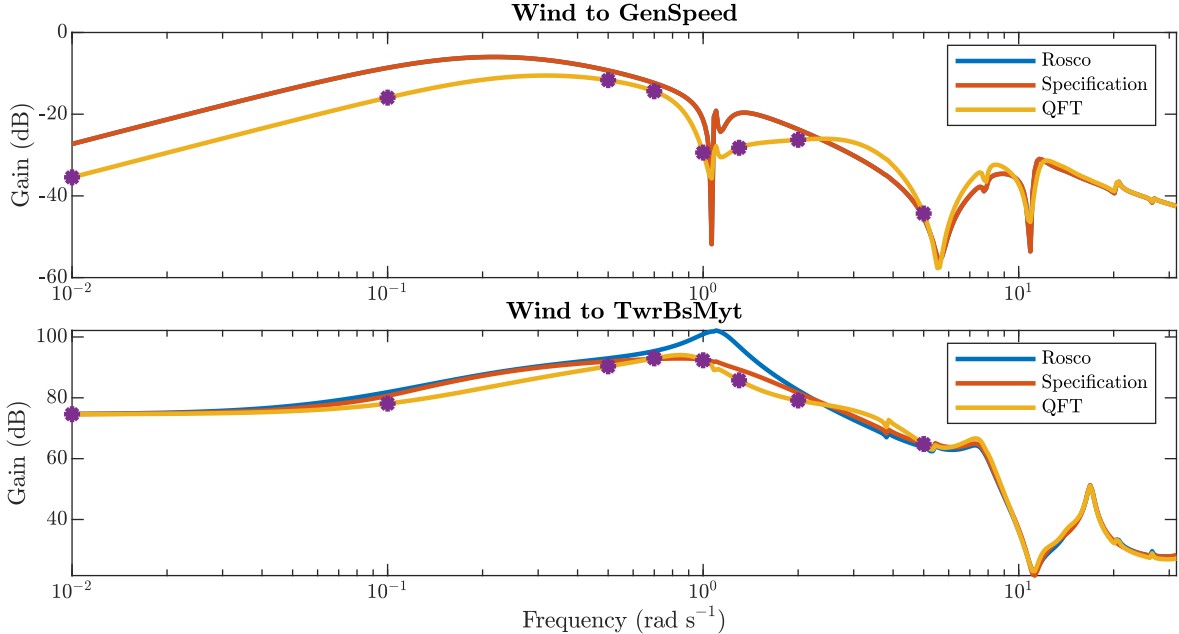

**Figure 7.** Magnitude Bode plot for ROSCO (blue), the design specifications (red) and the QFT controller (yellow) for the generator speed (upper plot) and the tower base load (lower plot). The purple dots show the design frequencies in which the specifications are always met. A simple control structure guarantees that the response between design frequencies is smooth and that specifications are meth most of the times.

### 4.4 Result analysis

Once a pitch controller and an ATD controller have been designed for each operating point in the above rated region (11 to 25 m/s), they have been integrated in the aeroelastic simulator using its Simulink interface. Due to the difference structure of the controllers at different operating points, an output blending strategy has been used for the interpolation between controllers. A filtered pitch signal has been used as interpolation variable.

At each operating point, four different seeds have been used for the generation of turbulent wind fields of length 600s and class B turbulent intensity. Wind fields have been generated using the tool Turbsim (Jonkman (2006)).



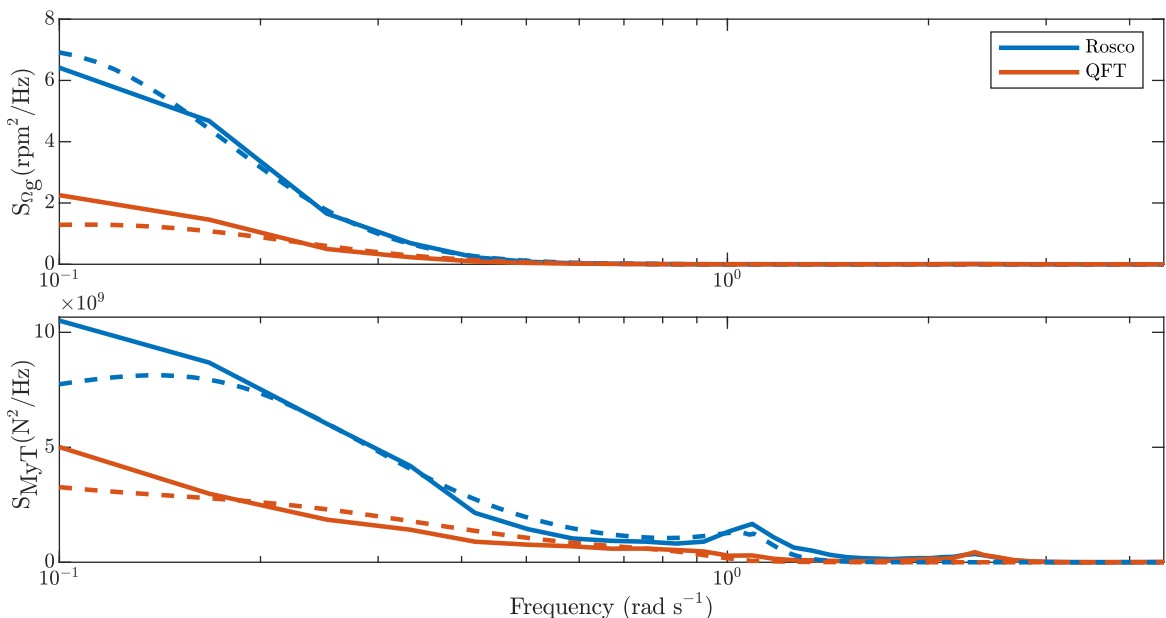

**Figure 8.** Validation of the correspondence between linear (dashed) and nonlinear (solid) models with the ROSCO (blue) and QFT (red) controllers. Besides a good matching between both models, the effect of the QFT controllers can be observed in the lower magnitude of the spectra, especially in frequencies below 0.5 rad/s and in the peak corresponding to the first fore-aft mode of the tower (close to 1 rad/s).

The first step in the analysis of the results consists on the validation of the relation between the linear and the nonlinear models. Figure 8 shows a comparison between the theoretical and simulation spectra of the generator speed and tower base bending moment with the baseline and the QFT controllers at an operating point of 19 m/s. The theoretical spectra have been obtained using the linear model of the wind turbine and the theoretical spectrum of rotor effective wind speed. The simulation spectra have been obtained by appliying Welch method to the time series obtained from OpenFAST. Two main conclusions

can be extracted from this figure: (i) the QFT controller outperforms ROSCO controller for both outputs and (ii) there exists a good correspondence between the estimation obtained from the linear model and the results of the nonlinear simulation. This second fact is critical for the performance of the proposed fatigue estimation method.

| Performance parameters | Operating point (m/s) | | | | | | |
|---|---|---|---|---|---|---|---|
| | 15 | 17 | 19 | 21 | 23 | 25 | Mean |
| Generator speed std | 31.54% | -1.91% | -39.48% | -48.96% | -50.07% | -52.18% | |
| Mean power | 0.47% | 0.37% | 0.05% | -0.17% | -0.42% | -0.46% | |
| Tower base DEL | -25.91% | -22.24% | -27.75% | -27.28% | -34.71% | -28.84% | |
| Pitch activity | -8.22% | -0.53% | 1.13% | 6.66% | 1.87% | 5.43% | |

**Table 2.** Comparison of the main performance indicators for the ROSCO and QFT controllers.





Even though improving the performance of ROSCO is not the main objective of this work, the performance of the QFT controller has been studied to understand the posibilites of this design technique and the fatigue based identification. The performance of the controllers has been measured with four main indicators. The standard deviation of the generator speed is used as a measure of the quality of the generator speed regulation. Table 2 shows how the standard deviation is reduced for all operating points but 15 m/s, which is closer to the transition between regions. The mean generated power shows small variations at different operating points that cancel each other when calculating the total average. As expected from the linear model prediction, the fatigue at the tower base is reduced significantly at all operating points. Lastly, pitch activity is evaluated using the standard deviation of the collective pitch signal, which is significantly reduced at 15 m/s and then increases for wind speeds 19 to 25 m/s.

Similar information can be perceived in Figure 9, which represents the simulation outputs for a single wind seed with mean speed 19 m/s. Both the generator speed and the tower base load show smaller deviations from their mean value for the QFT controller, which account for a smalled standard deviation and fatigue respectively. Power mean value is close to 15MW in both cases. The presence of the ATD controller in the QFT configuration can be observed in the ripple that appears in the pitch signal, accounting for an increased pitch standard deviation.

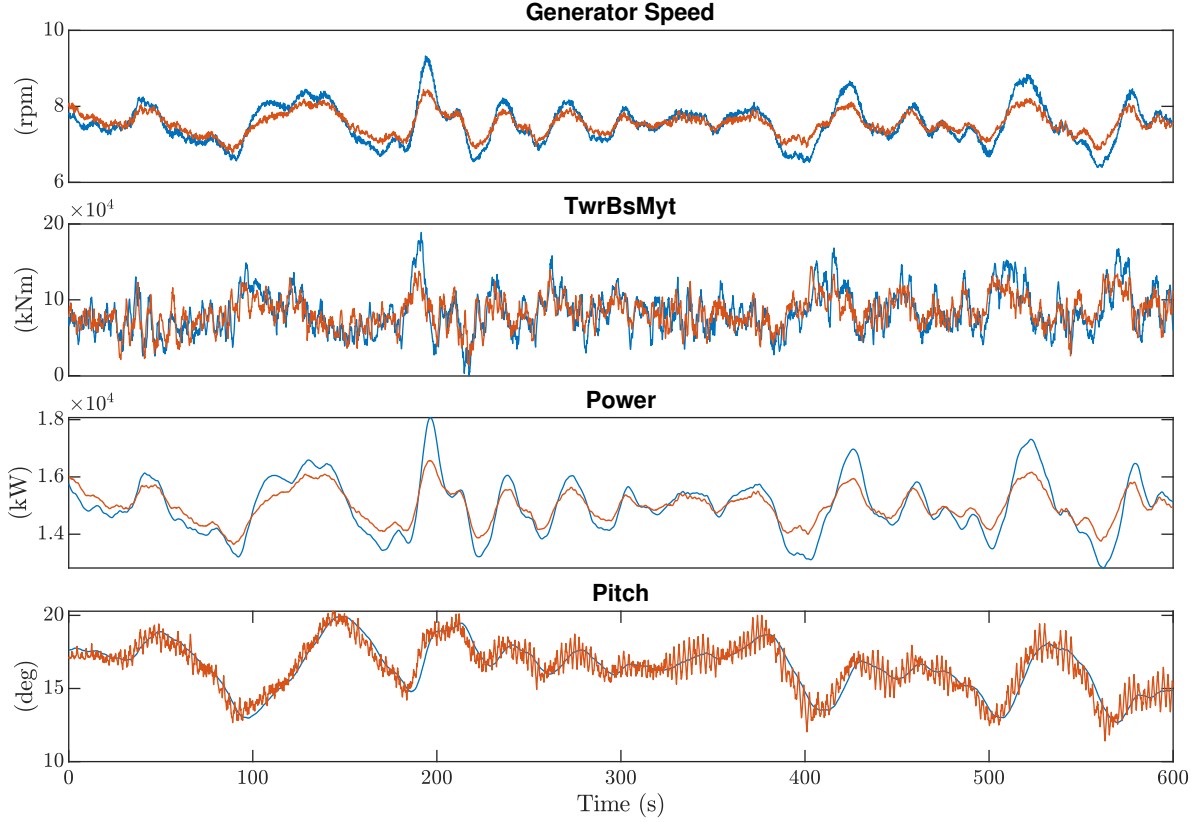

**Figure 9.** Time series of relevant variables in an OpenFAST simulation with mean wind speed 19 m/s. The standard deviation of generator speed and power is visibly smaller for the QFT controller (red) than for ROSCO (blue). Besides, the fatigue in the tower base load should





also be reduced as the ATD controller mitigates the bigger peaks in the blue line, that do not appear for the QFT controller. Lastly, the lower plot shows that the use of an ATD controller results on an increased high frequency pitch activity.

The main result of this work is the prediction of fatigue with the use of linear models. Table 4.4 shows the fatigue reduction promised by the specifications, the fatigue reduction estimation provided by the linear approximation and the actual fatigue
reduction calculated using a rainflow counting algorithm and the S-N curve. Two main conclusions can be extracted from the data: (i) the fatigue predicted by the linear model is always smaller than the promised by the specifications and (ii) the deviation between the linear prediction and the rainflow-based fatigue evaluation is smaller than 2% for all operating points, having an averave value of 1.07%.

| Fatigue estimation | Operating point (m/s) | | | | | | |
|---|---|---|---|---|---|---|---|
| | 15 | 17 | 19 | 21 | 23 | 25 | Mean |
| Specifications | 37.37% | 10.10% | 11.78% | 20.55% | 35.97% | 30.35% | |
| Linear prediction | 24.81% | 24.15% | 28.60% | 27.93% | 35.12% | 30.36% | |
| Rainflow evaluation | 25.91% | 22.24% | 27.75% | 27.28% | 34.71% | 28.84% | |
| Estimation deviation | 1.1% | 1.91% | 0.85% | 0.65% | 0.41% | 1.52% | 1.07% |

**Table 3.** Fatigue reduction at different operating points. The Table includes information on the fatigue reduction promised by the specification, the reduction estimated with the help of linear models and the actual fatigue reduction as evaluated with a rainflow counting algorithm.

Figure 10 shows a graphical representation of the accuracy of the linear approximation of fatigue by plotting the frequency-
domain estimation again the time-domain evaluation. The black line ($y = x$) serves as a reference of a perfect prediction, while the actual data is represented as a star per operating point. Appart from the point corresponding to 15m/s, the linear approximation tends to overstimate the fatigue reduction, although the error remains admisible for controller design purposes.





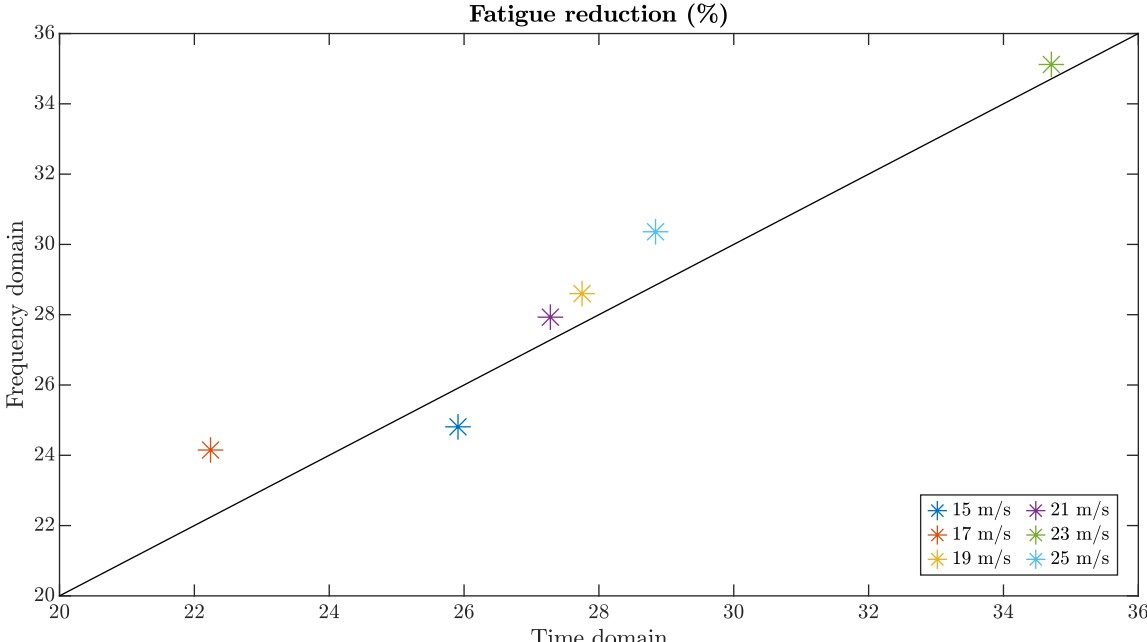

**Figure 10.** Graphical comparison of the linear fatigue estimation (frequency domain) and the rainflow-based fatigue assessment (time domain). The black line presents the perfect-estimation reference, while the starts represent the actual relation between the frequency domain and the time domain results for different operating points.

## 5 Conclusions

This work has presented a method for the estimation of mechanical fatigue based on the use of the linear model of a wind turbine and its controllers and a numerical calculation of the variation of the damaged with the load spectrum. Then, this method has been used for the design of control specifications.

The method has been validated with the design of pitch controllers for the 15MW reference wind turbine. More specifically, the feedback pitch controller and the active tower damping have been designed to improve the performance of the ROSCO

PI controller. With the aid of the linear fatigue estimation and the design of specifications, fatigue has been reduced by 22 to 36% at operating points ranging 15 to 25 m/s mean wind speed. Due to the good correspondence between the linear fatigue prediction and the fatigue evaluation obtained with a rainflow algorithm (under a 2%), a single iteration in the non-linear simulation step has been required. Consequently, the design process has been accelerated significantly.

The error present in the fatigue estimation can be attributed to two different factors. The first one is that the results provided

by the linear model do not exactly match the ones provided by the nonlinear simulator. Besides the numerous non-linearities present in the realistic model of the wind turbine and the complete control structure, other phenomena such as the 3P frequency and the spatial variation of the wind cannot be taken into account by a linear approximation. The second factor is the difference between time-domain and frequency-domain fatigue evaluation methods. The rainflow-counting algorithm has proven to be





the best approximation for fatigue estimation and is the reference in the standard. On the other side, due to the fundamental
empirical nature of frequency-domain methods, their performance varies significantly depending on the characteristics of the
load and the system. While Dirlik's method has proven to provide a good result, the search for a more accurate method remains
open.

All in all, the use of linear models for fatigue estimation has proven to be useful and their use can be extended. On the one
hand, the use of this frequency domain estimation of fatigue could be included in the iterative design of other wind turbine
components, such as structural elements. On the other hand, their use could also be studied for different controller design
methologies, the most obvious one being $H_\infty$. Lastly, the full potential of the methodology could be achieved for multi-
objective design, in which the role of the control system in the fatigue of different elements could be linked to the bussines
case of the wind turbine.

*Author contributions.* Irene Miquelez-Madariaga: Conception, simulation data collection, writing. Jesús Arellano: Conception, writing.
Daniel Lacheta-Lecumberri: Simulation data collection, proofreading. Jorge Elso: Conception, proofreading.

*Competing interests.* The contact author has declared that none of the authors has any competing interests.



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
