# Peer review of "Obtaining fatigue-based frequency domain specifications for the design of controllers in wind turbines"

_Wind Energy Science, 2024_

## Author Comment (AC1)

**REVIEWER 1**

**General comments**

This work aims to help close the gap between controller design in the frequency domain and fatigue assessment which is done using nonlinear time-domain simulations. The proposed method uses Dirlik's fatigue estimation based on linear models and uses numerical differentiation to estimate the sensitivity of fatigue to the closed-loop performance of the linear system. This sensitivity can then be used to inform the control design in two ways: Which frequencies are important for fatigue reduction and estimate the change in fatigue based on a change in the controller (specification). The method is applied to improve the fatigue alleviation of the collective pitch and active tower damper controller.

Running nonlinear time-domain simulations over multiple wind seeds, as the IEC standards prescribe for fatigue analysis, takes a long time and prohibits fast iterations in the controller design. This method can reduce the iteration time between controller design and (simplified) analysis before a full nonlinear analysis is run. This makes this work scientifically relevant. However, I see two main improvements for this manuscript:

- The objective of the method is not aligned between the title, abstract, introduction, method, validation, and conclusion. Reading the method and validation section, I understand that this method estimates fatigue based on a controller or controller specification. However, from the introduction and conclusion, it seems like the control designer can specify how much fatigue reduction is desired and, based on this, make a control specification to achieve this. It's a subtle difference, but it makes the whole manuscript sometimes difficult to follow.
- Dirlik's fatigue estimation method requires a regression to estimate the parameters G1, G2, and G3. However, it is not discussed how this regression is done. This requires (expensive) time-domain simulations to obtain data, making this slow. There is currently no discussion on the trade-off between first doing a slow calibration of Dirlik's method followed by fast control iterations (of which they only require a single iteration in this work) and the 'conventional' method of only using the slow time-domain simulations.

Overall, I think this work is suitable for wind energy science and could be accepted after revisions.

We want to thank the reviewer for the thorough and helpful review of our work. We find they took the time to try to understand and improve our article by adding helpful suggestions. Before going in to the more specific comments, we wanted to address the main improvements:

We see how the text might be confusing and have tried to simplify it and align the different sections. We have begun by changing the title to "Linear estimation of fatigue for the design and evaluation of controllers in wind turbines" and the main characteristics of the method in the abstract. Similar changes have been introduced through the text.

Regarding the use of Dirlik's method, we find the suggestion might be based on a poor writing of our original text. The values of G1, G2, and G3 can be found directly by applying the formulas developed by Dirlik. We just wanted to empathise the empirical nature of the method, but we did not mean that we have performed any data fitting to it.

**Specific comments**

I have summarized my interpretation of the objective from different sections, to highlight the misalignment.

- Title: The title signals that you can obtain controller specifications but the method shows that rather you can use controller specifications to generate a fatigue estimation, so the other way around.
- Abstract: "This work presents a methodology to (i) estimate fatigue using linear models and (ii) generate control specifications directly linked to the mechanical fatigue caused by driving loads for a wind turbine applications."
- I find 'linked' too vague, and (like was the case for the title) if I look at eq (9), you don't strictly generate control specifications based on fatigue but rather calculate the fatigue based on control specifications (similar to the title).
- Introduction: "The proposed method has two main characteristics: (1) it allows to identify the range of frequencies in which the specification contributes more to the overall fatigue and (2) it allows to quantify the expected fatigue decrease when a change in the specification or the controller is introduced."
- I think this is the clearest stated objective of the work that is also in line with the method and validation. Compared to the introduction, however, a new characteristic is introduced (identifying frequencies of interest).
- Method: The method section describes the method to estimate the sensitivity of the fatigue with respect to the output spectrum (eq 5) and the change in fatigue based on the control specification (eq 9).
- The validation section first uses eq (5) to identify frequencies of interest. I think it is then too vague about applying eq (9) to generate control specifications. The specification for desired fatigue reduction is only shown all the way at the end of this section.
- Conclusion: "This work has presented a method for the estimation of mechanical fatigue based on the use of the linear model of a wind turbine and its controllers and a numerical calculation of the variation of the damaged with the load spectrum. Then, this method has been used for the design of control specifications." This section again presents the objective in a slightly different way.

I think that aligning these different sections will greatly help readers in understanding the objective of the work and also in how the proposed method works and is applied. This will also make it much easier for readers to use your method in their own work. My recommendation would be to clearly define and give meaningful names to fatigue specification (percentage of desired fatigue reduction), control specification, linear prediction of fatigue based on the control specification, and linear prediction of fatigue based on the actual linear controller used in closed-loop. And subsequently clearly use these throughout the document and align the objectives stated in the different sections.

We have made several changes through the text, trying to follow your suggestion. Some of them are:

- A change in the title to "Linear estimation of fatigue for the design and evaluation of controllers in wind turbines".
- A change in the abstract to emphasise that the base of the method is the linear estimation of fatigue.
- A change in the introduction to emphasise that the method is based on the linear estimation of fatigue and its two main characteristics.
- Several more changes are included through the text. A special effort has been made to avoid synonyms and make the relation between concepts more clear.

**Introduction:**

- I think the argumentation is strong but could benefit by adding references to related scientific work.

We have added some more sources that we found pertinent such as Menezes (2018), Pegalajar-Jurado (2018) or Pao(2024).

- "This work" line 29: The first time reading this, it was unclear whether this referred to your work or to Tibaldi (2016).

  We have rephrased the sentence (lines 30-34):

  Some work has been done to solve this problem for a general wind turbine design application (Tibaldi (2016); Pao (2024)), but it has not targeted the design of control specifications. The main contribution of this article is a method that allows to design control specifications for a target fatigue reduction and to perform a fast evaluation of the obtained controller via a linear estimation of the fatigue.

**Frequency domain assessment of mechanical fatigue**

- Line 58: $\Delta D_{NB}$ is not defined in the text. The sentence introducing this equation says that "it estimates the number of stress cycles and their respective amplitudes..." making it seem like there are two outputs. By continuing the paper, it eventually becomes clear what it means, but it would be better if it was defined here.

  $\Delta D_{NB}$ is now explicitly introduced as "the damage caused by fatigue" in lines 58-59.

- I didn't know much about this subfield and feel that this provided a great overview.

  We are glad the section was useful.

**Using linear models for the estimation of fatigue and the generation of control specifications**

- Line 90-91: It would be nice to have a reference to literature talking about models of rotor effective wind speed.

  A paper by Soltani has been cited, where the concept of rotor effective wind is explained.

- Line 97: In these steps, I miss doing a regression to find G1, G2, and G3 for Dirlik's method that you use. It's fine if you want to specify that elsewhere if you don't consider it to be part of your main method. But also for the results section, it is unclear how you obtained G1, G2, and G3.

  We are afraid there has been a misunderstanding due to a wrong explanation on our side. Dirlik's method offers closed expressions for G1, G2 and G3, which depend on the spectral moments of the load. We wanted to state that the whole method was empirical, not we had done a regression to find the parameters. The whole explanation has been rephrased in lines 70-72 "where G1, G2, and G3 are empirical expressions depending on the spectral moments derived by Dirlik". The full expressions of G1, G2, and G3 have not been included in the article because they can easily be found in the literature.

- Line 100: Make it more specific by specifying 'the output spectrum'. In hindsight, it makes sense that it should be the output spectrum, but I think specifying it here already would just make it easier for readers to understand.

  We have added "the output spectrum" to make it more clear.

- Line 111: 'shows that higher frequencies lead to higher fatigue'. I don't see that as the highest frequencies don't have a high $\Delta D_\%$.

  We understand 'shows that higher frequencies lead to higher fatigue' is not correct and have replaced it by "higher frequencies contribute with more cycles than lower frequencies" which is what we intended to express. (Figure 1 caption).

- Line 116: 'which has been tested via simulation' is a slightly weak claim since you don't provide those results. But indeed, from your final results, we see that the estimation is quite accurate. Maybe you can make this claim stronger by providing these simulation results or referring forward to your results.

  A better explanation has been offered in lines 119 to 121: "The linearity in the relation between fatigue and the amplitude of the spectrum has been tested by changing the amplification or attenuation factor in the second step of the method. In all cases, the increase or decrease of fatigue was proportional to the chosen modification factor."

- Out of curiosity, why do you express $\Delta D$ and $\Delta S$ as percentages instead of absolute sensitivities/derivatives? If that is significant, it could be a nice insight to add to the method. (If not, also totally fine to leave out.)

  We do it because we are used to express desired fatigue reductions as percentages, but similar results would be expected if we worked with absolute sensitivities.

- Why did you choose to do a numerical differentiation of the fatigue damage rather than take the analytical derivative of Dirlik's fatigue formula? Again, if that is significant, it could be a nice insight to add to the method. (If not, also totally fine to leave out.)

  Coefficients G1, G2 and G3 are also functions of the spectral moments. As a consequence, the complete expression is a complex function of the spectral moments and finding an analytical derivative was not viable. As good results were validated with the numerical derivative, we did not pursue that path.

**Validation for the 15MW reference wind turbine**

- Line 138-139: ROSCO also has the option for tip speed ratio tracking control in below-rated, so the statement '...in which a quadratic control law is used...' can be more nuanced.

  A comment about the different options offered by ROSCO has been added in llines 144-145: " ROSCO offers different strategies for below rated operation, among which the quadratic control law has been chosen."

- Figure 2: It would be nice if the x-limits were the same between Figures 1, 2, and 4 for easier comparison as a reader.

  We have changed the limits in the figures.

- Line 190: Very nice explanation, but it might be stronger if you specify beforehand what the desired reduction of fatigue is. If I understand it correctly, your method can be used to know which frequencies are important for the control specification (this is also nicely illustrated here) _and_ to estimate a priori how much fatigue reduction you can expect, and design for. This latter point is not made in this section until Table 3 all the way at the

end of this section. I think you can show this aspect of your method by showing how you design the controller for a specific fatigue reduction.

Then intended fatigue reduction (15%) has been in line 204. Besides, it has been stated that specifications that improve that value are also valid.

- Line 197-199: You say that since the ROSCO specification is used, your controller is expected to outperform ROSCO. If the specification is equal to ROSCO, I would expect that they perform about the same. Why do you expect something else?

That is right, we have corrected our text (line 210) "This way, a performance similar to the baseline controller is expected in terms of generator speed regulation."

- Figures 5 and 6: There is no legend for the black line and the different circles. It would be good to add that.

Figures 5 and 6: I haven't used Nichols plots myself, so maybe this is easy to interpret for someone with more experience with them, but visually, it seems like there is too much going on, and it is difficult to see what the different lines do and represent. Do you think it would be possible to improve the visualization in these figures by decluttering them?

We understand these figures might be somewhat difficult to interpret for engineers without a background in control. However, as they are an exact representation of the design environment we see some value in keeping them as they are. To improve the readability of the document, we have added a description of the Nichols plot in lines 215 to 221: "lpshape is a graphical design tool that represents the bounds and the openloop nominal transfer functions in the Nichols plot (gain in dB agains phase in degrees). The bounds are the limits between the allowed and forbiden values for the open-loop nominal transfer function. They can have different colours, depending on their corresponding frequency, and solid and dashed lines indicating lower and upper limits. The open-loop nominal transfer function is represented by the solid black line and a set of circular markers at the design frequencies, with the same colour as 220 their corresponding bounds. The design is performed by modifying the position of the markers by adding zeros and poles to the controller."

- Table 2: The 'mean' column is empty.

We have calculated the mean values using a Weibull function. Lines 266-267: "Mean values have been obtained assuming a Weibull distribution in mean wind with shape factor of 2.2 and a scale factor of 11.29." We have also added the mean values to the table.

- Line 244-246 and Table 3: You use the standard deviation of the collective pitch signal to assess the pitch activity. I think this unfairly favors your method since the standard deviation doesn't penalize your higher frequency activity with the pitch actuators. I think you should use the actuator duty cycle, which is also more closely related to actuator wear.

We agree that a more thorough analysis of the cost of pitch action should be performed to obtain conclusive results. However, we find it out of the scope of our work. We have however added a disclaimer in lines 264-266: "Lastly, pitch activity is evaluated using the standard deviation of the collective pitch signal, which 265 is significantly reduced at 15 m/s and then increases for wind speeds 19 to 25 m/s. A more thorough analysis of the

cost of pitch control should be based on the actuator fatigue, where less favourable results would be expected.”

- Table 3: I think the paper would gain clarity when you consistently and throughout the whole paper refer to fatigue specification (percentage of desired fatigue reduction), control specification, linear prediction of fatigue based on the control specification, and linear prediction of fatigue based on the actual linear controller used in closed-loop. Maybe thinking of clear terms for these and defining them in the introduction or at the first mentioned place would help readers' understanding.

  We see how this might be a problem and have tried to simplify the text by avoiding synonyms and offering descriptions at the first appearance of each term.

- Figure 9: The legend of this figure is missing. Could you add one in addition to (or instead of) specifying the lines in the caption? This will also help colorblind people distinguish the lines.

  We have modified the figure and added a legend following your suggestion.

- Figure 10: This figure is also not colorblind friendly and uses quite a convoluted way to represent your data, in my opinion. I would suggest putting the wind speed on the x-axis, the fatigue reduction on the y-axis, and using colors to compare the frequency and time domain results.

  We have followed your suggestion by changing Figure 10.

- The acronym ATD is not defined on its first use.

  The acronym is now defined in line 146.

- It is unclear how the linearized models are obtained. Could you add a paragraph explaining how you obtained them?

  We have added an explanation in lines 158-159: “The open-loop linear models of the wind turbine have been obtained with the aid of OpenFASTs linearization tool and the closed-loop models have been calculated following the structure represented in Figure 3.”

- What is the turbulence intensity of the simulation?

  According the Normal Turbulence Model and the IEC61400, for a class B wind turbine the turbulence intensity changes depending on the mean wind speed. We have added more information on the topic in lines 245-247: “Each wind file has a 600 s length and a class B turbulent intensity according to the Normal Turbulence Model (NTM, Ishihara (2012)) and the IEC:61400:1-2019 standard. Wind fields have been generated using the tool Turbsim (Jonkman (2006)).”

**Conclusions**

- Is it fair to claim an improvement over the ROSCO controller without mentioning here again that the ROSCO controller does not have ATD?

  Not really. We have mentioned the absence of ATD in the systems description and again in the conclusions (line 263): As expected from the linear fatigue prediction, the fatigue

at the tower base is reduced significantly at all operating points, partly due to the absence of ATD in the baseline controller.

- Line 276: The sentence starting with 'Besides' doesn't make complete sense to me. 'Besides' signals a contradiction which is seemingly not present.

  We have removed the word besides, although to our understanding it expresses addition, not contradiction.

- Is another contributing factor to the difference between the actual and the estimated fatigue your assumption on linearity (on line 116)?

  It could be, although we consider it small. However, we now mention it in lines 304-305: "Lastly, the proposed method is based on a linear approximation to fatigue estimation, which can introduce an error."

**Technical corrections**

- Line 45: undefined citation
- Line 130: Spelling correction: System
- Line 158: Spelling correction: wind turbine
- Line 221: Spelling corrections: met most of the time
- Line 253: Referencing correction: Table 3
- Line 258: Spelling correction: table
- Line 266: Spelling correction: damage
- Line 287: Spelling correction: business

All spelling error have been corrected..

---

## Author Comment (AC2)

**REVIEWER 2**

**General comments:**

It is difficult to follow the whole paper. There are some miss match between the title, abstract, method, analysis, and conclusion. The method section shows how to estimate fatigue based on a controller specification/design. However, reading other part of the paper, it shows the control designer can specify the desired fatigue reduction when designing a controller, based on which a certain control specification can be achieved. This small miss match makes it difficult to read.

We want to thank the reviewer for the very useful comments on our work. We have included most of the suggestions in our text, as they have helped improving its quality. We understand that the objectives of the work and the properties of the method have been presented in slightly different ways through the text. We have tried to eliminate the mismatches and simplify the nomenclature.

Beside this, all the loads presented are only the tower base moments (fore-aft), which are used for showing the method works nicely. But, I am missing other loads, for example, what about the loads on blade root? what about the loads for the tower bottom side-side moments and the shaft yaw and tilt moments? It would be good to mention in order to have a complete picture.

Although we consider analysing more loads out of scope for the current article, we understand the relevance and intend to cover that topic in future work.

Additionally, Dirlik's fatigue estimation method requires a method to estimate the parameters (G1, G2, and G3). However, it is not discussed how this estimation is done.

We find this suggestion might be based on a poor writing of our original text. The values of G1, G2, and G3 can be found directly by applying the formulas developed by Dirlik. We just wanted to empathise the empirical nature of the method, but we did not mean that we have performed any data fitting to it.

The paper need to be major revised before it can be accepted.

**Specific comments:**

1. What is the meaning of $\Delta DNB$ in Equation (1), citing a text book or a paper may help the reader to understand it.

$\Delta D_{NB}$ is now explicitly introduced as "the damage caused by fatigue" in lines 58-59. It has been obtained from Wirsching(1980), which has been cited in line 55.

2. In Equation (3), you have mentioned the Transfer function T between input (Wind) and the generator speed ($\Omega_g$, but usually, wind is considered as a disturbance rather an input to a system. You also used in line 88, "disturbance input", so more percisely, in equation (3), it should be disturbance to the system.

In the context of control systems, both control actions and disturbances are considered inputs to the system. However, for more clarity, line 88 has been modified from "disturbance input" to "input".

3. Please cite a paper or text book to specify "rotor effective wind speed"

A paper by Soltani has been cited, where the concept of rotor effective wind is explained.

4. Line 100, the step 2, it would be eaiser to understand by the reader if you describe a bit more detailed on how to achieve, for example, 1% reduction on the "spectrum" or based on my understanding it is the "output spectrum" at a single frequency.

The reduction is achieved by simple multiplication, as the spectrum is not an analytical function. An explanation is added in lines 101-102.

5. Line 101, how do you compute new fatigue estimation $DOmega 0$ using eqation (2)? You did not mention how do you get G1, G2 and G3.

Dirlik's method offers closed expressions for G1, G2 and G3, which depend on the spectral moments of the load. The expressions appear in the cited references, but have been omitted to improve the readability of the text. Explicitly describing them would entail adding several new equations and parameters to the text that are not fundamental to the results and would be distracting. The full expression for Dirlik's fatigue approximation can easily be found in the literature.

6. The lower plot of Figure 1 shows the fatigue sensitivity ($Delta D\%$) as a function of frequency, it only shows that higher frequency results in smaller fatigue sensitivity. But, it does not show what you have mentioned: "higher frequencies lead to higher fatigue". Please explain this statement, because, it would affect the results or the conclusion of your paper.

We understand 'shows that higher frequencies lead to higher fatigue' is not correct and have replaced it by "higher frequencies contribute with more cycles than lower frequencies" which is what we intended to express. (Figure 1 caption).

7. Line 112, "QFT" it is the first time showing this abbrevation, you need to specify it, althought, it is quite obvious for the control engineer to know it refers to "quantitative feedback theory", But for other wind turbine engineers, it is not that obvious.

We have added the full name in line 114.

8. Line 146 You are saying: "wind speed (lower plot)", but the lower plot is not the wind speee, rather "the magnitude o fthe feedback controller". Please comment on this.

We have changed the sentence to make clear that the lower plot refers to the gain of the PI controller (lines 152-153): "Figure 2 shows the magnitude plot of the PI controller (lower plot) operating at a 19 m/s wind speed and its effect on the generator speed and tower base load."

9. In your context, you did not mention Figure 3, so what is the purpose of showing "Figure 3"?

Figure 3 was necessary to understand how closed-loop linear models have been obtained. This piece of information had been forgotten in the first version but has now been included, making the figure relevant.

10. Finally, at Line 152, you mentioned "Quantitative feedback theory" as "QFT", it should be mentioned before when you use "QFT" in the first time.

We have now fixed the mistake by introducing Quantitative Feedback theory before.

11. Line 166, "open-loop transfer function"? I think it should be "close-loop transfer function".

No, bounds are defined for the open-loop transfer function. The loopshaping is also based on the open-loop transfer function.

12. Line 174, "some skills", could you be more precise? what skills?

Skills has been substituted by practice (line 185), which conveys the intended meaning in a more precise way.

13. Line 178, why do yo use four seeds per wind speed? it is fine to say "reducing number of simulations, etc" or cite other paper says, it is enough to use 4 seeds.

To our experience, 4 seeds are enough for an evaluation in a controller design context. Thorough evaluations that are more thorough are typically performed by loads engineers.

14. In general, you should insert a space between number and their units, eg. "19 m/s" rather than, "19m/s".

We have corrected the mistake through the text.

15. Line 185, I don't see there is a large peak at 2.3 rad/s in Figure 1. it is only a small peak, I would say the large peak is at round 1 rad/s, which seems to be the 1st tower mode.

The lower plot of Figure 4 shows a small peak at frequency 2.3 rad/s. Although it is definitely smaller than the first fore-aft mode of the tower, it will contribute to fatigue as well.

16. Now, you mention in Line 190: "Even though the lower frequencies have the greatest impact on the fatigue (see the lower plot in Figure 4) ", which is an counter statement comparing the one that you made for Figure 1. If I compare Figure 1 and 4, they are very similar but you have completely different statements.

We understand 'shows that higher frequencies lead to higher fatigue' is not correct and have replaced it by "higher frequencies contribute with more cycles than lower frequencies" which is what we intended to express. (Figure 1 caption).

17. Line 197: You are saying: "the ROSCO specification is used" for the design of your controller. I would expect that the perform of your controller should be the same as ROSCO controller. So, Why do you expect it is better?

We cannot expect it to be better, we have corrected our text (line 210) "This way, a performance similar to the baseline controller is expected in terms of generator speed regulation."

18. Figure 5 and 6 are very difficult for a non control engineer to understand. Do you think you can use a better representation?

We understand these figures might be somewhat difficult to interpret for engineers without a background in control. However, as they are an exact representation of the design environment we see some value in keeping them as they are. To improve the readability of the document, we have added a description of the Nichols plot in lines 215 to 221: "lpshape is a graphical design tool that represents the bounds and the openloop nominal transfer functions in the Nichols plot (gain in dB agains phase in degrees). The bounds are the limits between the allowed and forbiden values for the open-loop nominal transfer function. They can have different colours, depending on their corresponding frequency, and solid and dashed lines indicating lower and upper limits. The open-loop nominal transfer function is represented by the solid black line and a set of circular

markers at the design frequencies, with the same colour as 220 their corresponding bounds. The design is performed by modifying the position of the markers by adding zeros and poles to the controller."

19. In the whole paper, I don't see the description of your linearized model in term of Equation, etc, and I don't see how do you get the linearized model. You need to add a section on describing your linearized model and how to obtain it.

We have added an explanation in lines 158-159: "The open-loop linear models of the wind turbine have been obtained with the aid of OpenFASTs linearization tool and the closed-loop models have been calculated following the structure represented in Figure 3."

**Comments for correcting some writing mistakes:**

- Line 45: missing citation
- Line 47: controller's -> controller
- Line 88: 2 time "the disturbance", please remove one.
- Line 130: Sistem -> System
- Line 158: win turbine  -> wind turbine
- Line 163: is -> are
- Line 213: sorrounding -> surrounding
- Line 221: meth -> met
- Line 249: smalled -> small
- Line 253: There is no Table 4.4
- Line 258: averave -> average
- Line 259: Table -> table
- Line 260: again -> against
- Line 266: damaged -> damage
- Line 287: bussines -> business
- Table 2, the "mean" column is empty.

All spelling error have been corrected. We have calculated the mean values using a Weibull function. Lines 266-267: "Mean values have been obtained assuming a Weibull distribution in mean wind with shape factor of 2.2 and a scale factor of 11.29." We have also added the mean values to the table.